# REFACTORBENCH: EVALUATING STATEFUL REASONING IN LANGUAGE AGENTS THROUGH CODE

**Dhruv Gautam**[1]* **Spandan Garg**[2] **Jinu Jang**[2] **Neel Sundaresan**[2]
**Roshanak Zilouchian Moghaddam**[2]
[1]UC Berkeley  [2]Microsoft

## ABSTRACT

Recent advances in language model (LM) agents and function calling have enabled autonomous, feedback-driven systems to solve problems across various digital domains. To better understand the unique limitations of LM agents, we introduce RefactorBench, a benchmark consisting of 100 large handcrafted multi-file refactoring tasks in popular open-source repositories. Solving tasks within RefactorBench requires thorough exploration of dependencies across multiple files and strong adherence to relevant instructions. Every task is defined by 3 natural language instructions of varying specificity and is mutually exclusive, allowing for the creation of longer combined tasks on the same repository. Baselines on RefactorBench reveal that current LM agents struggle with simple compositional tasks, solving only 22% of tasks with base instructions, in contrast to a human developer with short time constraints solving 87%. Through trajectory analysis, we identify various unique failure modes of LM agents, and further explore the failure mode of tracking past actions. By adapting a baseline agent to condition on representations of state, we achieve a 43.9% improvement in solving RefactorBench tasks. We further extend our state-aware approach to encompass entire digital environments and outline potential directions for future research. RefactorBench aims to support the study of LM agents by providing a set of real-world, multi-hop tasks within the realm of code.[1]

## 1 INTRODUCTION

*"Repetition is the root of all software evil" — Martin Fowler*

Large language models (LLMs) have been quickly acquiring new capabilities (Bubeck et al., 2023), leading towards adoption of AI-powered systems in various formats and domains. The increasing usage of LLM powered tools like Github Copilot have greatly improved the capability of developers in software development tasks (Peng et al., 2023). More recently, an emphasis on multi-step execution through LLM feedback loops has unlocked the ability to solve harder problems within a variety of fields (Reed et al., 2022; Sumers et al., 2024; Yao & Narasimhan, 2023), including parts of software engineering.

This new paradigm of solving larger software tasks has led to the construction of a variety of new automated software engineering (ASE) systems, most being structured as LM agents (Wang et al., 2024c; Cognition.ai, 2024; AWS Q Developer, 2024; Gauthier, 2024; Aide.dev, 2024; Örwall, 2024; Yang et al., 2024; Tufano et al., 2024; Wang et al., 2024d; Chen et al., 2024a; Zhang et al., 2024b; Arora et al., 2024; Xia et al., 2024; Zhang et al., 2024a). Evaluations for such systems are currently largely comprised from real world data on Github (Jimenez et al., 2024; LaBash et al., 2024). While being the strongest open-source signal for software engineering tasks at scale, Github is inherently noisy through its snapshot nature, also requiring strong filtration and validation testing for reliable evaluations (Chowdhury et al., 2024; Bowman & Dahl, 2021). We find that the filtration causes skewed task styles, creating a necessity for new data to diversify coding benchmarks.

---

*Work done during internship at Microsoft. Correspondence to dhruvgautam@berkeley.edu

[1]Data available at: https://github.com/microsoft/RefactorBench

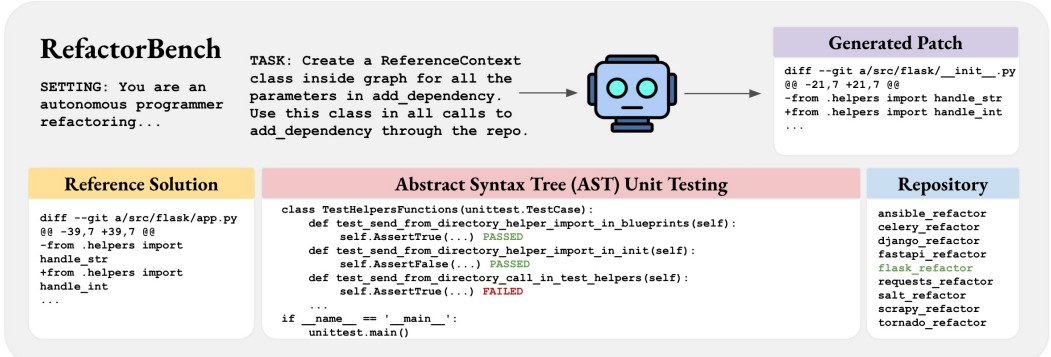

Figure 1: RefactorBench task instances are multi-file refactors verified by custom AST unit testing. Tasks are split over popular open-source Python repositories and reference solutions are withheld to prevent overfitting to the task.

To address these challenges, we present RefactorBench, a benchmark designed to evaluate the largely undocumented task of multi-file code refactoring in large codebases. Unlike isolated function-level edits, multi-file refactoring requires comprehensive reasoning and composition of multiple smaller changes. Our benchmark, RefactorBench, also allows for controlled analysis into instruction-following through multiple instruction sets with specified and unspecified objectives. As LLMs have been extremely proficient in function level editing over model generations (Jiang et al., 2024), we find it important to evaluate the abilities of general LM agents given that they can reliably perform core subtasks, which we verify in RefactorBench's thorough filtering process. With unique abstract syntax tree (AST) based unit testing, the evaluation suite checks for a comprehensive variety of subtasks necessitated by the core refactor without dependence on exact line match.

Through evaluations of a baseline agent on RefactorBench, we find overfitting and poor performance, solving a maximum of 35% of tasks with our easiest instruction set. We also see a variety of unique failure modes, many centering around LM agents struggling to track and reason about previous actions. Similarly, extensive work in policy learning has commonly faced issues in long horizon execution (Piterbarg et al., 2023; Chen et al., 2024b; Hejna et al., 2023). By editing agent interfaces, we explore introducing conditioning over state updates, a common tactic in neural agent design, to our real world language agents and see 71% increases in subtask completion rates.

Overall, our contributions in this work are threefold:

1. We introduce RefactorBench, a benchmark of code refactoring tasks that necessitate edits in multiple files and reasoning based on previous actions taken.

2. We evaluate multiple open-source systems on RefactorBench and analyze *three novel failure modes* isolated through differing baseline runs.

3. We construct *state-aware interfaces* and show improvement in the reasoning capabilities of a modified baseline agent.

## 2 BACKGROUND

### 2.1 RELATED WORK

**Benchmarks** SWE-bench, a benchmark consisting of GitHub issues, is the community standard for evaluating open-ended problem-solving in code environments (Jimenez et al., 2024). Our work, comparably, focuses on handcrafted and underrepresented multi-file refactoring tasks, isolating unique language agent behaviors. Unlike existing function-level code benchmarks (Chen et al., 2021; Austin et al., 2021), which also include refactors, we concentrate on the challenges posed by multi-file edits. Through a lazy and descriptive instruction set to accompany base instructions, we also build on previous works that scale evaluations of LLMs' instruction-following capabilities with lazier instructions (Cassano et al., 2024). Recent works have also focused on evaluating

repository-level code completion systems (Liu et al., 2024a; Agarwal et al., 2024; Bairi et al., 2023), but our work differs by evaluating larger actions than exact line match, using generalist evaluations through AST testing. New works have also started to benchmark LLMs across the life cycles of various engineering tasks (Li et al., 2024; Huang et al., 2024; Xie et al., 2024c), and recent LM agent benchmarks have also started to evaluate for planning, reasoning, and decision-making abilities in multi-turn generation settings (Liu et al., 2023b; Xie et al., 2024b). We combine these two threads by evaluating on engineering tasks that necessitate multi-turn actions. Moreover, newer benchmarks have shown that emulating differing environments can help identify unique failure modes (Ruan et al., 2024; Yao et al., 2024). Similarly, in RefactorBench, we find that multi-file dependencies in code provide a strong test bed for previously unseen failure modes in LM agents.

**Compositional Tasks and Memory** Some benchmarks have identified modeling long-term dependencies as a difficult task for core LMs (Tay et al., 2021; Xie et al., 2024b; Wang et al., 2023b; Lee et al., 2024a). To combat this issue, many works have targeted changes in model architecture and training (Gu et al., 2022; Liu et al., 2023a; Gu & Dao, 2024; Xiong et al., 2024). Other works have tackled this style of problem through augmenting LM agents to have external memory in order to learn longer term skills after large sequences of actions (Sarch et al., 2023; Shinn et al., 2023; Wang et al., 2023a; Sumers et al., 2024). Differing from long term memory in language agents, we largely focus on enabling concurrent reasoning and state-aware behaviors in language agents. Comparable to this focus, many works in embodied control and small neural agents have previously explored training and conditioning over observations about current state in multi-turn situations (Chen et al., 2024b; Wang et al., 2024a; Piterbarg et al., 2024; Li et al., 2022; Moreno et al., 2021). We explore extending these concepts to real-world LM agents to improve their performance.

## 2.2 DEFINITIONS

We generalize varied perspectives in previous literature to construct our definition of a language/LM agent and related terms: a core LM receives an user instruction $u$ and executes actions $a_n$ using a set of tools $t_m$, receiving partial observations $\omega_n$. This follows a structure most similar to a partially observable Markov decision process (POMDP) (Kaelbling et al., 1998), where the trajectory is $\tau_N = (a_1, \omega_1, \dots, a_N, \omega_N)$. This largely matches the formalization of LM agents articulated in ToolEmu (Ruan et al., 2024). We also define and use the words *state* or *stateful* in the context of LM agents as the nature of being dependent on the accumulation of actions $a$ and observations $\omega$, though not necessarily all generated from the LM. Importantly, *stateful reasoning* focuses on making decisions based on the current state, which is partially observable and can change dynamically. Previous works in building LM agents have recognized the importance of designing interfaces that allow the core LMs to make better decisions for a variety of tasks (Yang et al., 2024; Liu et al., 2024b; Wang et al., 2024d; Lu et al., 2024; Shang et al., 2024). We reference the design choices behind $t_m$ and it's impact on $\omega_n$, $a_n$ as *interface design*.

## 3 REFACTORBENCH

RefactorBench is benchmark of handcrafted multi-file refactoring tasks. The goal for each task is to generate a patch that changes the repository to follow the rules of a specified refactor. In this section, we describe our end-to-end process of constructing the refactoring tasks and highlight some important features of RefactorBench.

### 3.1 TASK CONSTRUCTION

To design a benchmark capturing the common practice of code refactoring, we focus on including a diversity of styles of tasks, using Fowler et al. (2018) as a reference point for different styles of refactors. As the test beds for all tasks, we first select 9 popular Python repositories that have differing overall file structures (Table 1). We then run the below four step process on each repository:

**Step 1: Localization and Filtering.** We leverage LLMs to identify potential refactoring opportunities in repositories. We iteratively prompt gpt-4o (OpenAI, 2024) with complete files from a target repository, along with examples of various refactor types from Fowler et al. (2018), requesting line numbers and suggestions for potential refactors. We then filter through the returned sites manually

to verify if corresponding suggestions can be made and if the changes would affect multiple files. This process yields a list of refactoring suggestions and their corresponding edit locations in each repository.

**Step 2: Construction of Reference Solutions.** To generate a prospective reference solution for each refactor, a group of experienced Python programmers handcraft unique, related edits to the refactoring suggestions generated in the previous step. These edits are made based on the design principle (Fowler et al., 2018), while concurrently using `gpt-4o` to verify that each core refactor is tractable by the language model. Tractability verification is done through prompting `gpt-4o` with the file to edit, the design principle, and a summary of the change needed to be made. We define this process in depth in Section 3.2.

**Step 3: Development of Testing Files.** Once the tractability of the subtasks are verified, the developers then create relevant unit tests for each overarching task. At minimum, for every core edit verified in Step 2, a new unit test is generated that parses through the respective file's AST and verifies that changes have the correct broad code structure and syntax necessitated by that subtask, removing dependence on exact-match testing (Appendix D). This iterative approach creates a breadth of tests that comprises a necessary minimum for the total refactor. At test time, a LM agent's generated solution is applied to the codebase and the associated tests crafted for the task instance are then executed. *A generated patch is considered successful, if all of the relevant AST tests pass.*

**Step 4: Generation of Relevant Task Instructions.** After reference solution and AST test creation, the developers write a short, but comprehensive task summary to help in the instruction design phase, what we refer to as the *base instruction*. In order to evaluate different degrees of instruction following with specified and unspecified objectives, we generate two other instruction sets: *lazy instruction* and *descriptive instruction*. These instruction sets are generated through a few-shot learning prompt with the respective base instruction and the related unit tests (Appendix A.1).

The above four step process yielded 100 large overarching multi-file refactoring tasks and corresponding tests in 9 different Python repositories. Throughout this work, we report the success on a run based on passing all tests for a task. Table 1 and Figure 2 show a breakdown by repository and other statistics related to the tasks.

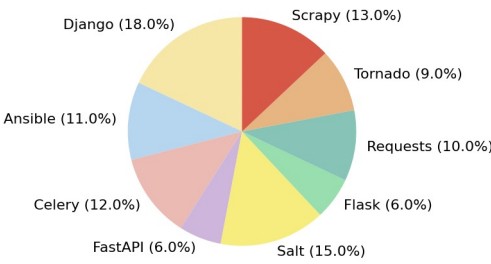

Figure 2: Distribution of RefactorBench tasks across 9 open source Python repositories.

Table 1: Statistics on RefactorBench tasks, repositories, and AST-based unit tests.

|  |  | Mean | Max |
|---|---|---|---|
| Lazy Instruction | Length (Words) | 16.0 | 38 |
| Base Instruction | Length (Words) | 20.6 | 44 |
| Desc. Instruction | Length (Words) | 68.8 | 116 |
| Codebase | # Files | 2327.6 | 6815 |
|  | # Lines | 304K | 1.8M |
| Reference Solution | # Files edited | 4.3 | 31 |
| AST Tests | # Length (Functions) | 6.5 | 27 |
|  | # Length (Lines) | 131.1 | 378 |

## 3.2 IMPORTANT FEATURES

**Multi-File** By filtering out single-file refactors as part of our task construction process, all tasks in RefactorBench involve multi-file edits. Our tasks edit between 2 to 31 files, with 4 files edited in our reference solutions on average. This feature, by definition, detracts the ability of single-shot LLMs of solving the tasks, and forces feedback-based editing systems to reason over multiple files.

**Varying Instruction Sets** RefactorBench offers three sets of instructions with varying degree of descriptiveness. With multiple instruction sets, we are able to test for a breadth of types of instruction-following and provide a way to effectively scale the difficulty of RefactorBench. The lazy instructions match the styles of real users, where objectives are often unspecific. We also include the base instruction which describes the task completely in a succinct manner. And through the descriptive instruction, we are able to evaluate on an exhaustive instruction where systems are given insights on what they will be tested on, a theoretical upper bound on performance.

**Subtask AST Testing**  In RefactorBench, unit tests for each task are designed to cover various subtasks the LM agent needs to accomplish. During the test construction process, we separate the unit tests to break apart the behavior of subtasks within tasks. This makes understanding the failures within patches an interpretable and quick process. For instance, one can evaluate which files the agent makes edits in, giving more comprehensive understanding of the order of tasks and proximity to a correct solution. RefactorBench's unit tests comprise of 2 to 27 subtests, with an average of 6.51 tests per task. See Appendix D for an example test file and Appendix E for multiple test outputs seen through the lens of this subtask testing format on RefactorBench.

**Tractability**  Through verification steps during task construction, we also make sure that all the core edits are feasible by frontier models at the time of writing. Due to this, our refactors have stronger signal on evaluating the reasoning behaviors between function calls of LLM feedback loops rather than the broad open-ended task of generating passing code changes. Similarly, previous work has also recognized the importance of focusing agent benchmarks to interpretable subtasks (Côté et al., 2019; Xie et al., 2024b; Shridhar et al., 2021; Chowdhury et al., 2024). Overall, this tractability requirement allows for a more dedicated focus on evaluating the stateful reasoning abilities of LM agents.

## 4 EXPERIMENTS

In this section, we explain our approaches to evaluate language agents on RefactorBench. All main studies are done on SWE-agent, which is the highest performing open-source agent framework on the full SWE-bench split at the time of writing. SWE-agent also structurally follows our earlier definition of a POMDP-based LM agent (Yang et al., 2024; Jimenez et al., 2024), while other agents sample from multiple agents (Wang et al., 2024d), weakening ablation studies. Often, due to costs and rate limits on model endpoints impacting efficient ablation studies, we opt to use `gpt-4` in experiments, but find that our results scale similarly across models.

> **Update:** These baselines are outdated compared to current SWE-agent implementations. We share more updated baselines in Section 7.3.

### 4.1 PRELIMINARIES

**Current systems have overfit to solving reproducible bugs.**  As a prior, we observed poor performance for some LM agents when running them on simpler tasks in RefactorBench. Upon investigating internal code of a few open-source LM agents, we find that their internal prompting and in-context examples steer towards solving Github issues. This task specific prompting causes these language agents to treat refactoring problems as bug-fixes. For instance, many systems will attempt to create a bug reproduction script for a simple renaming task. We causate this initial finding as a result of having a lack of benchmarks: *it is hard to robustify LM agent systems without ways to evaluate on diverse styles of tasks* (Kapoor et al., 2024; Dehghani et al., 2021). For the rest of this work, to better understand the frontier of capabilities within current systems, we alter internal prompts to focus on the task of refactoring. We therefore consider these baselines as an upper bound on performance, and hope for future systems to be designed in accordance to and evaluate over diverse styles of problems. We discuss directions for future systems and generalist performance in Section 7.

### 4.2 BASELINES

Using a containerized framework that emulates a user file system with the target repository, we run a baseline of SWE-agent on all RefactorBench tasks with a per instance cost limit of $10.00[2]. We report the percentage of completely successful task instances on each run. On the lazy, base, and descriptive instruction sets, SWE-agent with `gpt-4` solves 12%, 18%, and 27% respectively. To verify generalization across models, we also run the descriptive baseline with `claude-3.5-sonnet`, which solves 35% of the test cases completely. To contextualize this performance, we have a proficient human developer attempt all the tasks within the benchmark, with a limit of 5 minutes per task

---

[2]This cost is due to large token counts being necessitated with multi-file reasoning.

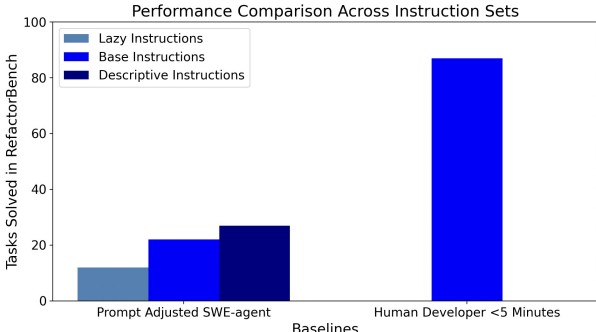

Figure 3: Baselines of a prompt adjusted SWE-agent with `gpt-4` and a human developer given IDE access and a time limit of 5 minutes.

Table 2: Baseline task performance relative to instruction type. Through the varied categories, we find that language agents are sensitive to unspecified objectives (Lazy) and improve in performance greatly when given information on which files to make edits in (Descriptive).

| Instruction Type | Resolution Rate |
| --- | --- |
| Lazy | 12.0% |
| Base | 22.0% |
| **Descriptive** | **27.0%** |

using the base instructions, and they solve 87% of the test cases. The average length of a successful trajectory using `gpt-4` is about 45.8 actions and the overall average length is about 58.5 actions.

Additionally, we sample 3 random *solved* RefactorBench instances in repositories that have 3 or more solved, and combine their descriptive prompts to run as singular instances. We find that SWE-agent, although able to solve the singular tasks, is unable to solve any of these longer pseudotasks. We further discuss related results and tackle long horizon planning in later sections of this work.

## 5 ANALYSIS

From manual review of trajectories on RefactorBench, we find repeating general behaviors language agents perform. Many prior works have outlined some strengths and failures of current LM agents in different scenarios (Yang et al., 2024; Wang et al., 2024d; Xie et al., 2024b). As such, we focus on three novel failure modes isolated through our baseline experiments. After large-scale human review of trajectories and developing an understanding of failures, to confirm their prevalence on a larger scale, we use `gpt-4` with reference solution diffs to analyze unresolved trajectories and the respective test outputs as following one of the failure modes in this section. Through this, we classify about 58% of failed trajectories are corresponding to one of these failure modes, and in held out validation, a human reviewer agreed with the classifications about 74% of the time.

**Agents fail to find relevant locations and make applicable changes.** Through our task construction, our descriptive instructions provide information on all files that need to be edited. However, we still observe through about 44% of the tests checking for some change, agents initialized with the descriptive instruction did not edit the target files, although being prompted to. These results *differ* from previous results that firmly found that most LM agent coding systems create patches at the correct location, and mainly fail through incorrect implementations (Chen et al., 2024a; Yang et al., 2024). Moreover, none of the tasks that require changes in more than 6 files are solved in any of our baselines. These results complement previous work evaluating planning capabilities of LLMs, where increases in constraints correlate with decreases in performance (Xie et al., 2024b; Huang et al., 2022). We hypothesize that the increase in files serves as a proxy constraint and LM agents fail in both efficient exploration and composing previous actions. We formalize the related problems of action tracking and stateful reasoning in-depth and tackle it through state updates in Section 6.

**Agents fail due to interactions that necessitate erroneous intermediate states.** Our classifications also show that 78.4% of trajectories error in a code editing step. Through analyzing these trajectories, we commonly encounter cases where making a change that temporarily introduces errors is a necessary step to solve the task. This is often because subsequent modifications, either within the same file or across multiple files, are concurrently required to resolve these issues. Consequently, the practice of automatically enforcing strict linting rules and rejecting edits based on errors proves to be an impractical approach for scaling real world agents, even though most open-source systems have previously found in-edit linting to significantly boost performance for bug fixing (Örwall, 2024; Yang et al., 2024; Wang et al., 2024d). This identified scenario demonstrates that LM agents often

imitate human forms of interaction, and removing innate capabilities through guardrails can backfire in unintended manners. We further discuss unobstructed LM agent interaction in Section 7.1.

**Agents fail due to context flooding and losing sight of objectives.** We find that agents struggle in decision making after having commands that are rejected due to formatting issues or unexpected output (Figure 8). In recent work defining in-context reward hacking (ICRH) (Pan et al., 2024), LLMs, through feedback loops in small synthetic tasks, have been shown to model proxy objectives when optimizing over some larger objective. We find, in our real world task of refactoring, that the negative effects from ICRH are also accentuated by extensive context space being taken up by the handling of constraint violations, deprioritizing the initial objective in a form of few-shot learning (Brown et al., 2020). Specifically, a common linting error edit in our tasks shows the model an average of about 1,466 tokens [3]: comprising of two blocks of code, error handling prompts, and the flagged errors. We find that this lengthy repetition for error handling function calling *weakens trajectory structure*. For instance, in some trajectories, we see the agent run end after an intensive function level feedback loop is resolved, a form of prioritizing the new objective. Language models losing sight of initial goals has recently been tackled within single-shot code generation tasks through attention dilution (Tian & Zhang, 2024), but we find this new issue is more prevalent in LM agent trajectories, and is exacerbated through the context-expensive feedback loops. We further discuss ways to approach robust trajectory reasoning in Section 7.1.

## 6    TOWARDS STATE-AWARE LANGUAGE AGENTS

In our analysis, we found a general issue with LM agents struggling to plan edits in the right locations. We hypothesize that an innate limitation of the POMDP setup of LM agents is that after sufficiently many timesteps $n$, due to partial observability, the LM's understanding of the current state at such action $a_n$ becomes weaker, through divergence from the initial state before $a_1$. In this section, we first explore this claim through a synthetic setup and then attempt conditioning over state updates to improve on the failure mode. We later generalize our approach to entire digital environments and discuss the implications of state-awareness for agent interaction.

**Computation of state almost linearly decreases with respect to number of actions taken.** Similar to previous work in testing entity tracking in language models (Kim & Schuster, 2023), we test the existence of the earlier divergence hypothesis in a synthetic setup through prompting a LM with 15 categories of preferences, emulating a web agent. We iteratively give an increasing list of $n$ updates to the preferences (i.e. Dogs to Dislikes) and prompt the LM to output the updated list of preferences. Based on 125 randomly initialized runs, we find that failures in cumulative state construction linearly scale with the amount of actions taken (Figure 4). We show the exact setup for reproducibility in Appendix E.

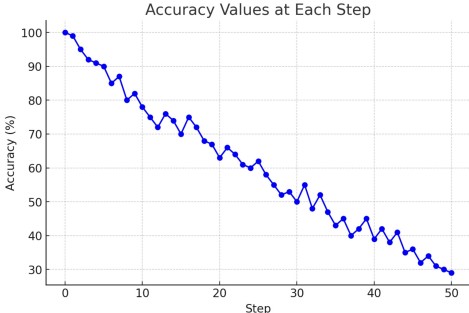
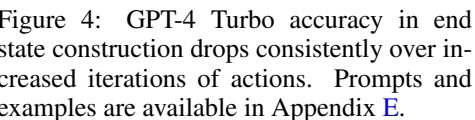

Figure 4: GPT-4 Turbo accuracy in end state construction drops consistently over increased iterations of actions. Prompts and examples are available in Appendix E.

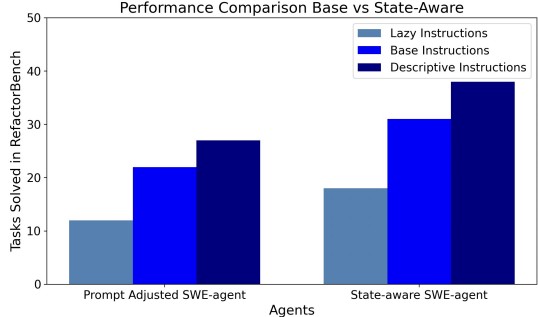

Figure 5: Comparison of a prompt adjusted SWE-agent and a state-aware SWE-agent implementation, both using `gpt-4`. Sample implementation code available in Appendix G.

---

[3]Using the o200k_base tokenizer on task instances within the `ansible/ansible` repository

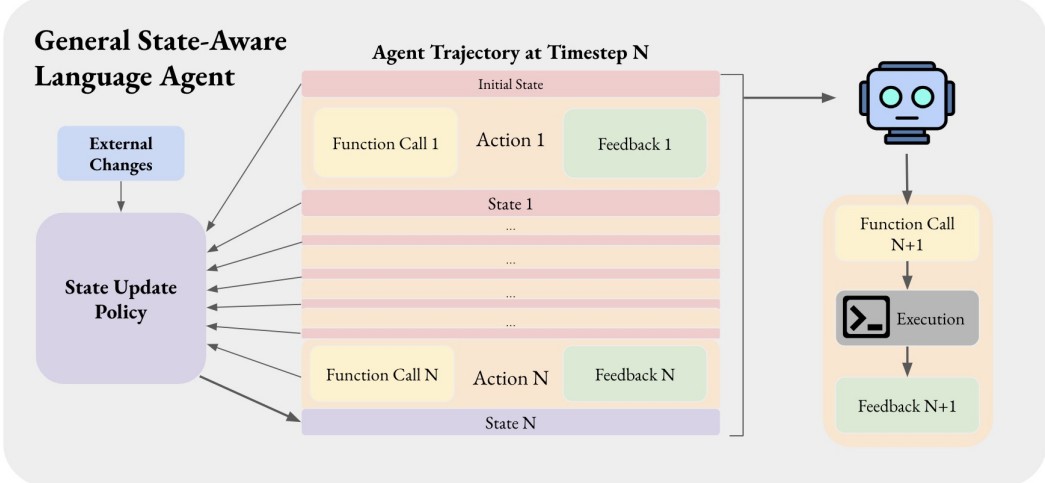

Figure 6: Example flow of a language agent at some timestep N interacting with a state update policy to generate the natural language state summary appended to the agent trajectory. The new updated trajectory is passed to the core LM to generate the next function call and execute it.

## 6.1 STATE-AWARE INTERFACES

In our baseline runs, we find that agent trajectories extend long (60+), necessitating actions across multiple files. However, real world implementations of LM agents often restrict the amount of previous information $\text{Count}(\omega)$ in $\tau_N$ to a controlled number of steps[4] to avoid flooding context windows. Being able to model long term changes with limited context has been a problem space in neural policy learning (Piterbarg et al., 2023). To tackle this, a recent SOTA approach in NetHack, a long horizon video game requiring continual learning (Hambro et al., 2022), used unix `diff` on previous observations in order to keep base models on track (Piterbarg et al., 2024). Their results confirm the importance of continuous and efficient modeling of state changes in environments, but also demonstrate that `diff` history exploits structure that is present apriori in observations. Other works using LMs to plan for embodied systems have found computation of state observations alongside baseline observations important for long-horizon task planning (Chen et al., 2024b). We combine the idea of efficient modeling of state observations with previous proven results with feedback-based interface design (Yang et al., 2024; Shang et al., 2024), to motivate our approach: *state-aware interfaces*.

Our implementation for a state-aware interface for interacting with code focuses on succinctly representing the divergence from initialization state, which is represented through previous edit actions. As such, before every function call, we have a cached and updated section with information related to all previous edits, prompting the model with an understanding of the accumulation of its own changes (Appendix G). Formally, we add a recurring externally computed state variable $\sigma$ to our POMDP, where $\sigma_N$ is the state at timestep N, and our trajectory now follows $\tau_N = (a_1, \omega_1, \sigma_1, \ldots, a_N, \omega_N, \sigma_N)$.

**Agents with state updates have stronger performance in RefactorBench tasks.** We modify SWE-agent with `gpt-4o` to track and display representations of state (Appendix G). This change boosts the agent's overall performance on RefactorBench: an average of 43.9% relative increase over the instruction sets compared to baseline agents (Figure 5). We also find a strong upwards trend in subtask completion: an average of 71% increase over the instruction sets. As our abstract syntax tree testing isolates unique subtasks in different files and functions, we find that passing more subtasks is correlated with stronger stateful reasoning, the intended goal of the state-aware interface.

---

[4]Five step observational window in the case of SWE-agent.

## 6.2 STATE UPDATE POLICIES

Having precomputation from the state update allows the LM to ignore computing the reconstruction subtask when generating the next function call. We extend state-updates to generalize to entire digital environments, not just an agent, through the construction of a *state update policy*.

We define our state update policy $\pi_{\text{state}}$ as a function that manages and provides the cumulative state information to the agent within an environment. Formally, let $\sigma_N$ represent the current state at timestep $N$ derived from all prior interactions. The state update policy $\pi_{\text{state}}$ can be expressed as a conditional function:

$$\pi_{\text{state}} : (\tau_N, \sigma_{N-1}, ..., \sigma_0) \to \sigma_N,$$

where $\tau_N = (a_1, \omega_1, \sigma_1, \ldots, a_{N-1}, \omega_{N-1}, \sigma_{N-1}, a_N, \omega_N)$ is the trajectory up to the generation of $\sigma_N$, including all actions $a_i$, observations $\omega_i$, and prior states $\sigma_i$.

**State update policies can lead to deviations from typical sequential agent reasoning.** We find states expressed in natural language to be a natural approach to facilitate concurrent interactions between language agents in open digital environments. Through our initial implementation of a state update policy, we are also able to model simple external changes from a concurrent state-aware user (Figure 6). In a simple example, we concurrently make a change with an external agent to rename a function that the LM agent has already edited to complete it's refactoring task. Through the state update policy, we are able to propagate this edit information and agent is able to decide to later view the new edit for more context (Appendix H). However, as all the tasks in RefactorBench are mutually exclusive, we do not further evaluate on modeling conflicting objectives between agents at a larger scale, but expand on similar directions for future work in Section 7.1.

## 7 DISCUSSION

We introduce RefactorBench, a benchmark that isolates unique failure modes of LM agents through code. Through our experiments, we find that most agents struggle at composing simple actions, and a diverse set of task evaluations is necessary for understanding and designing generalist language agent systems. We also show improvement on baselines through natural language representations of state and hope that further studies within stateful reasoning in differing scenarios can aid in the a larger understanding of the limitations of language agents.

### 7.1 FUTURE DIRECTIONS

Although there are many avenues to take for improving LM agents, we generalize our analysis from our evaluations on RefactorBench tasks into two main categories.

**Reasoning** Through the synthetic state construction experiment, we formalize that language models innately lose state understanding with respect to actions taken. As such, alongside our introduction of state update policies, we hypothesize that constructing smarter ways to generalize context rather than having the LM condition over the full trajectory is an important direction for tackling this problem. Various recent works on gist-based memory systems within agents, collaboration through optimizable graphs, exploration methods, skill learning, and mitigating partial observability seem promising (Zhuge et al., 2024; Nayak et al., 2024; Lee et al., 2024b; Wang et al., 2023a; Bruce et al., 2023; Xie et al., 2024a; Allen et al., 2024), but no works have tackled concrete methods to scale concurrent state-awareness for simple agent tasks. Many new approaches to improve agent performance have also been shown to scale up inference compute and score higher on various agent related benchmarks (Zhang et al., 2024a; Kapoor et al., 2024; Brown et al., 2024; Wang et al., 2024b). As real world refactoring results are not immediately verifiable, we find this style of repeated sampling to be insufficient without robust critic models. We encourage future works to scale inference time compute in language agents with open-ended tasks like those in RefactorBench.

**Interaction** In regards to interaction with the real world, we find that LM agents edit code in inefficient manners and have low success rates per single edit. Many agents have switched to diff-based editing (Örwall, 2024; Gauthier, 2024), which has empirically shown to be a more scalable solution. However, these systems do not get around the issues that come with temporary error states (Section 5) and format restrictions. The natural approach of full-file edits has its own distinct

issues: such as generalizing for files longer than token limits, inference speed, token cost, and context flooding. Future approaches could attempt to intertwine full-file rewrites with speculative decoding (Cursor, 2024) and custom trajectory truncation schemes to limit context window flooding. Overall, even outside of code generation, we predict this interaction problem for language agents to be of importance in varying digital domains, and we expect interaction to be a large focus in generalist agent interface constructions, especially in multi-agent scenarios. Our state update policy demonstrates a primitive case of agents being aware of other actions, and we hope for future works to generalize the environment-specific policy approach (Figure 6) in a variety of digital tasks.

## 7.2 LIMITATIONS

RefactorBench's task instances are all in Python, and we hope to expand the benchmark to various languages that are statically and dynamically typed, allowing for evaluations on more styles of refactors. We also focus on highly used open-source Python repositories, and language models may have a better understanding of the repositories due to their prevalence in training data. RefactorBench also has a limited amount of task instances due to the intensive process to create a singular end-to-end task and the necessity for quick evaluations (evaluations still takes hours with RefactorBench). In our evaluations, we also raise the cost limit much past limits in previous works in software engineering agents, due to the inability for agents to solve multi-file tasks quickly and cheaply. We also find, similar to previous works, that agent runs are not deterministic and can solve differing tasks in different runs. RefactorBench is a step forward in evaluating LM agents in robust manners through complex task construction, but like all benchmarks, is still plagued by the possible issues of over-fit data distributions (i.e. only refactors) (Kapoor et al., 2024). To prevent this repetitive issue, we do not release gold reference solutions (only the testing files) and we recommend evaluating software engineering agents on multiple styles of tasks: function editing tasks, bug fixes in SWE-bench, refactors in RefactorBench, etc. to truly define robustness in a general coding agent. Creating a general multi-faceted evaluation suite for language models and agents interacting with code is a compelling direction for future work.

## 7.3 UPDATED BASELINES

We evaluate the updated base SWE-agent 1.0 on RefactorBench using `gpt-4o`. We largely find that our previous results hold, with SWE-agent solving 21% of base instances and 31% of descriptive instances. However, we find that the speed and cost in which SWE-agent completes tasks significantly decreases to $1.69 per successful instance, showcasing new improvements in agents localizing changes to make. We are excited to see the community test their autonomous systems on RefactorBench, especially with reasoning models. Similar to recent results on SWE-bench, we have seen preliminary results of reasoning agents solving more RefactorBench tasks compared to non-reasoning model agents.

### ACKNOWLEDGMENTS

We thank the entire Data&AI Group at Microsoft for feedback and valuable discussions throughout the work. Dhruv Gautam thanks Michele Tufano, Alexander Pan, and Joe Guan for feedback of an earlier draft of the paper and for many formative research discussions about language models, reasoning, and code generation.

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

# A    TASK CONSTRUCTION PROMPTS

## A.1    PROMPT FOR LAZY INSTRUCTION

We prompt `gpt-4-turbo` with the handcrafted base instruction based on all the edits and this prompt to get our lazy instruction.

```
Please convert the following instruction to be less specific. Do not
change the behavior of the task, but give a short, less descriptive
version of the task in human-like prose. Your final instruction should be
 a partial sentence and should not instruct to run any tests. It should
just describe the changes to the repository. Do not output ANYTHING ELSE
BUT THE NEW INSTRUCTION. Here is the original instruction:

{base_instruction}

Here are examples of lazy instructions:

{few_shot_lazy}

Remember to only output the NEW LAZY INSTRUCTION CORRESPONDING TO THE
BASE TASK.
```

## A.2    PROMPT FOR DESCRIPTIVE INSTRUCTION

We prompt `gpt-4-turbo` with the handcrafted base instruction based on all the edits, the corresponding testing file, and this prompt to get our descriptive instruction.

```
Please convert the following instruction to be more specific and have
specific filenames for edits (not paths). Do not change the behavior of
the task, but give a longer, more descriptive version of the task in
human-like specifications. Reason over the AST tests provided to give
more information on which files could be relevant, but do not give exact
implementation details or anything related to what generalizations the
tests are looking for. Your final instruction should be around 2-3 full
sentences and should not say to run any tests or anything like that. It
should just describe the changes to the repository. Do not output
ANYTHING ELSE BUT THE NEW INSTRUCTION. Here is the original instruction
and its related test file:

{base_instruction}

Test File Starts Here:

{inst_test_file}

End of Test File.

Here are examples of descriptive instructions:

{few_shot_desc}

Remember to only output the NEW DESCRIPTIVE INSTRUCTION CORRESPONDING TO
THE BASE TASK.
```

## B AGENT PROMPT CHANGES

### B.1 UPDATED SWE-AGENT SYSTEM PROMPT

As described in Section 4, we alter the SWE-agent prompt to stop the agent from creating bug reproduction scripts for refactors and focus on the style of task at hand. We also remove pytest and other testing functionality in order to prevent timeout issues with the large codebases.

```
SETTING: You are an autonomous programmer specializing in refactoring,
and you're working directly in the command line with a special interface.
The special interface consists of a file editor that shows you WINDOW
lines of a file at a time.
In addition to typical bash commands, you can also use the following
commands to help you navigate and edit files.

COMMANDS:
command\_docs

Please note that THE EDIT COMMAND REQUIRES PROPER INDENTATION.
If you'd like to add the line ' print(x)' you must fully write that out,
with all those spaces before the code! Indentation is important and code
that is not indented correctly will fail and require fixing before it can
 be run.

RESPONSE FORMAT:
Your shell prompt is formatted as follows:
(Open file: <path>) <cwd>

You need to format your output using two fields: discussion and command.
Your output should always include \_one\_ discussion and \_one\_ command
field EXACTLY as in the following example:

DISCUSSION
First I'll start by using ls to see what files are in the current
directory. Then maybe we can look at some relevant files to see what they
 look like.
\begin{verbatim}
ls -a
\end{verbatim}

You should only include a *SINGLE* command in the command section and
then wait for a response from the shell before continuing with more
discussion and commands. Everything you include in the DISCUSSION section
 will be saved for future reference.
If you'd like to issue two commands at once, PLEASE DO NOT DO THAT!
Please instead first submit just the first command, and then after
receiving a response you'll be able to issue the second command.
You're free to use any other bash commands you want (e.g. find, grep, cat
, ls, cd) in addition to the special commands listed above.
However, the environment does NOT support interactive session commands (e
.g. python, vim), so please do not invoke them.

instance\_template: |-
We're currently solving the following issue within our repository. Here's
 the issue text:
ISSUE:

INSTRUCTIONS:
Now, you're going to solve this refactoring issue on your own. Your
terminal session has started and you're in the repository's root
directory. You can use any bash commands or the special interface to help
 you. Edit all the files you need to and run any checks or tests that you
 want.
Remember, YOU CAN ONLY ENTER ONE COMMAND AT A TIME. You should always
wait for feedback after every command.
```

When you're satisfied with all of the changes you've made, you can submit
 your changes to the code base by simply running the submit command.
Note however that you cannot use any interactive session commands (e.g.
python, vim) in this environment, but you can write scripts and run them.
 E.g. you can write a python script and then run it with `python <script\
_name>.py`.

NOTE ABOUT THE EDIT COMMAND: Indentation really matters! When editing a
file, make sure to insert appropriate indentation before each line!

IMPORTANT TIPS:
1. Always start by checking your working directory, cd'ing to the task
repo, and then trying to find where to do the refactor using the search
tools. Do not go into other directories like root or sys. Just go to the
task repo and make edits in there.

2. If you run a command and it doesn't work, try running a different
command. A command that did not work once will not work the second time
unless you modify it!

3. If you open a file and need to get to an area around a specific line
that is not in the first 100 lines, say line 583, don't just use the
scroll\_down command multiple times. Instead, use the goto 583 command.
It's much quicker.

4. Always make sure to look at the currently open file and the current
working directory (which appears right after the currently open file).
The currently open file might be in a different directory than the
working directory! Note that some commands, such as 'create', open files,
 so they might change the current open file.

5. When editing files, it is easy to accidentally specify a wrong line
number or to write code with incorrect indentation. Always check the code
 after you issue an edit to make sure that it reflects what you wanted to
 accomplish. If it didn't, issue another command to fix it.

6. It may be necessary to install the repository from source before you
can run code. Please think about how to install the environment from the
repository directory if you need to do so.

7. Do not run the repo tests, just focus on comprehensively finishing the
 multi-file refactor to the best of your abilities.

8. If you make a change that could have dependencies, always reason and
search where imports need to be updated and update them accordingly.

9. Do not run huge grep commands that could print out and flood the
context with too many items. If not sure, just check how many items the
command will run to begin with.

(Open file: open\_file) \\
(Current directory: working\_dir) \\

## C    EXAMPLE CONDENSED SUCCESSFUL TRAJECTORY

We show a portion of a successful trajectory for the refactoring task: rename-send-from-directory in the flask repository. Full examples are available in our repository.

Base Instruction: Rename the helper function send_from_directory to send_from_directory_helper but keep the overall flask usage as send_from_directory.

```
State-aware SWE-agent Run
{
  "action": "ls -F\n",
  "observation": "flask_refactor/\nbin@\nboot/\ndev/\netc/\nhome/\nlib@
  \nmedia/\nmnt/\nopt/\nproc/\nroot/\nrun/\nsbin@\nsrv/\nsys/\ntmp/\
  nusr/\nvar/",
  "response": "DISCUSSION\nFirst, I'll list the files in the current
  directory to get an overview of the project structure. This will help
   me locate the 'helpers' module and the 'flask' folder where the
  changes need to be made.\n```\nls -F\n```",
  "state": "{\"working_dir\": \"flask_refactor\", \"open_file\": \"n/a
  \", \"recent_edits\": []}",
  "thought": "DISCUSSION\nFirst, I'll list the files in the current
  directory to get an overview of the project structure. This will help
   me locate the 'helpers' module and the 'flask' folder where the
  changes need to be made.\n"
},
...
... (Makes changes in multiple files)
...
{
  "role": "assistant",
  "content": "DISCUSSION\nWe need to update the import statement for `
  send_from_directory` to import `send_from_directory_helper` as `
  send_from_directory`. Let's make the necessary edit.\n```\nedit 9:9\
  nfrom flask.helpers import send_from_directory_helper as
  send_from_directory\nend_of_edit\n```",
  "thought": "DISCUSSION\nWe need to update the import statement for `
  send_from_directory` to import `send_from_directory_helper` as `
  send_from_directory`. Let's make the necessary edit.\n",
  "action": "edit 9:9\nfrom flask.helpers import
  send_from_directory_helper as send_from_directory\nend_of_edit\n",
},
{
  "role": "user",
  "content": "[Truncated File Output]
  (Current State: ['Edited helpers.py at lines 514:514', 'Edited
  __init__.py at lines 24:24', 'Edited app.py at lines 42:42', 'Edited
  blueprints.py at lines 9:9', 'Edited test_helpers.py at lines 9:9\\n
  '])
  (Open file: /flask_refactor/tests/test_helpers.py)
  (Current directory: flask_refactor)
  bash-$",
},
{
  "role": "assistant",
  "content": "DISCUSSION\nThe import statement in `test_helpers.py` has
   been updated. We have now made all the necessary changes to the
  relevant files. Let's submit the changes to the code base.\n```\
  nsubmit\n```",
  "thought": "DISCUSSION\nThe import statement in `test_helpers.py` has
   been updated. We have now made all the necessary changes to the
  relevant files. Let's submit the changes to the code base.\n",
  "action": "submit\n",
}
```

## D  EXAMPLE TEST FILE

Here is the parameterize-gunzip.py test file for the parameterize-gunzip task:

**Base instruction: Encapsulate the parameters in gunzip into a new class called GunzipParams and update all references within the repository to create a GunzipParams object.**

```python
import unittest
import os
import ast

class TestScrapyMigration(unittest.TestCase):

    def test_gunzipparams_class_exists(self):
        # Path to the file where the class should be defined
        file_path = '../scrapy/utils/gz.py'

        # Check if the file exists
        self.assertTrue(os.path.exists(file_path), f"{file_path} does not
        exist")

        # Check if the GunzipParams class is defined in gz.py
        with open(file_path, 'r') as file:
            tree = ast.parse(file.read())

        class_found = False
        for node in ast.walk(tree):
            if isinstance(node, ast.ClassDef) and node.name == 'GunzipParams
            ':
                class_found = True
                break

        self.assertTrue(class_found, "Class 'GunzipParams' not found in gz.
        py")

    def test_gunzipparams_has_data_and_max_size(self):
        # Path to the file where the class should be defined
        file_path = '../scrapy/utils/gz.py'

        # Check if the file exists
        self.assertTrue(os.path.exists(file_path), f"{file_path} does not
        exist")

        # Check if the GunzipParams class has self.data and self.max_size
        attributes
        with open(file_path, 'r') as file:
            tree = ast.parse(file.read())

        class_node = None
        for node in ast.walk(tree):
            if isinstance(node, ast.ClassDef) and node.name == 'GunzipParams
            ':
                class_node = node
                break

        self.assertIsNotNone(class_node, "Class 'GunzipParams' not found in
         gz.py")

        data_found = False
        max_size_found = False
        for node in ast.walk(class_node):
            if isinstance(node, ast.Assign):
                for target in node.targets:
                    if isinstance(target, ast.Attribute) and target.attr == '
                    data':
```

```
                data_found = True
            if isinstance(target, ast.Attribute) and target.attr == '
            max_size':
                max_size_found = True

    self.assertTrue(data_found, "Attribute 'self.data' not found in
    GunzipParams class")
    self.assertTrue(max_size_found, "Attribute 'self.max_size' not
    found in GunzipParams class")

def test_gunzip_function_signature(self):
    # Path to the file where the function should be defined
    file_path = '../scrapy/utils/gz.py'

    # Check if the file exists
    self.assertTrue(os.path.exists(file_path), f"{file_path} does not
    exist")

    # Check if the gunzip function has the correct signature
    with open(file_path, 'r') as file:
        tree = ast.parse(file.read())

    function_found = False
    for node in ast.walk(tree):
        if isinstance(node, ast.FunctionDef) and node.name == 'gunzip':
            # Check function parameters
            args = node.args
            if len(args.args) == 1 and isinstance(args.args[0].annotation
            , ast.Name) and args.args[0].annotation.id == 'GunzipParams':
                # Check return type
                if isinstance(node.returns, ast.Name) and node.returns.id
                == 'bytes':
                    function_found = True
                    break

    self.assertTrue(function_found, "Function 'gunzip' with signature '
    def gunzip(params: GunzipParams) -> bytes' not found in gz.py")

def test_gunzip_in_sitemapspider(self):
    # Path to the file where SitemapSpider should be defined
    file_path = '../scrapy/spiders/sitemap.py'

    # Check if the file exists
    self.assertTrue(os.path.exists(file_path), f"{file_path} does not
    exist")

    # Check if the SitemapSpider class has a method _get_sitemap_body
    that uses gunzip with GunzipParams
    with open(file_path, 'r') as file:
        tree = ast.parse(file.read())

    sitemapspider_class = None
    for node in ast.walk(tree):
        if isinstance(node, ast.ClassDef) and node.name == '
        SitemapSpider':
            sitemapspider_class = node
            break

    self.assertIsNotNone(sitemapspider_class, "Class 'SitemapSpider'
    not found in sitemap.py")

    method_found = False
    gunzip_params_used = False
    for node in ast.walk(sitemapspider_class):
```

```python
            if isinstance(node, ast.FunctionDef) and node.name == '
            _get_sitemap_body':
                method_found = True
                for inner_node in ast.walk(node):
                    if isinstance(inner_node, ast.Call) and isinstance(
                    inner_node.func, ast.Name) and inner_node.func.id == '
                    gunzip':
                        if len(inner_node.args) == 1:
                            arg = inner_node.args[0]
                            # Check if the argument passed to gunzip is an
                            instance of GunzipParams
                            if isinstance(arg, ast.Name) or (isinstance(arg, ast
                            .Attribute) and arg.attr == 'GunzipParams'):
                                gunzip_params_used = True
                                break

        self.assertTrue(method_found, "Method '_get_sitemap_body' not found
         in SitemapSpider class")
        self.assertTrue(gunzip_params_used, "gunzip function inside '
        _get_sitemap_body' does not use a 'GunzipParams' object as a
        parameter")

    def test_imports_in_sitemap(self):
        # Path to the file where the imports should be defined
        file_path = '../scrapy/spiders/sitemap.py'

        # Check if the file exists
        self.assertTrue(os.path.exists(file_path), f"{file_path} does not
        exist")

        # Check if the correct import statement is present
        with open(file_path, 'r') as file:
            tree = ast.parse(file.read())

        imports_found = {
            "GunzipParams": False,
            "gunzip": False,
            "gzip_magic_number": False
        }

        for node in ast.walk(tree):
            if isinstance(node, ast.ImportFrom) and node.module == 'scrapy.
            utils.gz':
                for alias in node.names:
                    if alias.name in imports_found:
                        imports_found[alias.name] = True

        for import_name, found in imports_found.items():
            self.assertTrue(found, f"Import '{import_name}' not found in
            sitemap.py")

    def test_imports_in_test_utils_gz(self):
        # Path to the test file where the imports should be defined
        test_file_path = '../tests/test_utils_gz.py'

        # Check if the test file exists
        self.assertTrue(os.path.exists(test_file_path), f"{test_file_path}
        does not exist")

        # Check if the correct import statement is present
        with open(test_file_path, 'r') as file:
            tree = ast.parse(file.read())

        imports_found = {
            "GunzipParams": False,
```

```
        "gunzip": False,
        "gzip_magic_number": False
    }

    for node in ast.walk(tree):
        if isinstance(node, ast.ImportFrom) and node.module == 'scrapy.
        utils.gz':
            for alias in node.names:
                if alias.name in imports_found:
                    imports_found[alias.name] = True

    for import_name, found in imports_found.items():
        self.assertTrue(found, f"Import '{import_name}' not found in
        test_utils_gz.py")

def test_gunzipparams_used_in_test_utils_gz(self):
    # Path to the test file where gunzip should be used with
    GunzipParams
    test_file_path = '../tests/test_utils_gz.py'

    # Check if the test file exists
    self.assertTrue(os.path.exists(test_file_path), f"{test_file_path}
    does not exist")

    # Check if the gunzip function is used with GunzipParams in the
    test file
    with open(test_file_path, 'r') as file:
        tree = ast.parse(file.read())

    gunzip_params_used = False
    for node in ast.walk(tree):
        if isinstance(node, ast.Call) and isinstance(node.func, ast.Name
        ) and node.func.id == 'gunzip':
            if len(node.args) == 1:
                arg = node.args[0]
                # Check if the argument passed to gunzip is an instance of
                 GunzipParams
                if isinstance(arg, ast.Name) or (isinstance(arg, ast.
                Attribute) and arg.attr == 'GunzipParams'):
                    gunzip_params_used = True
                    break

    self.assertTrue(gunzip_params_used, "gunzip function in '
    test_utils_gz.py' does not use a 'GunzipParams' object as a
    parameter")

def test_imports_in_test_downloadermiddleware_httpcompression(self):
    # Path to the test file where the imports should be defined
    test_file_path = '../tests/
    test_downloadermiddleware_httpcompression.py'

    # Check if the test file exists
    self.assertTrue(os.path.exists(test_file_path), f"{test_file_path}
    does not exist")

    # Check if the correct import statement is present
    with open(test_file_path, 'r') as file:
        tree = ast.parse(file.read())

    imports_found = {
        "GunzipParams": False,
        "gunzip": False
    }

    for node in ast.walk(tree):
```

```python
        if isinstance(node, ast.ImportFrom) and node.module == 'scrapy.
        utils.gz':
            for alias in node.names:
                if alias.name in imports_found:
                    imports_found[alias.name] = True

    for import_name, found in imports_found.items():
        self.assertTrue(found, f"Import '{import_name}' not found in
        test_downloadermiddleware_httpcompression.py")

def test_gunzipparams_used_in_httpcompression_middleware(self):
    # Path to the middleware file where gunzip should be used with
    GunzipParams
    middleware_file_path = '../scrapy/downloadermiddlewares/
    httpcompression.py'

    # Check if the middleware file exists
    self.assertTrue(os.path.exists(middleware_file_path), f"{
    middleware_file_path} does not exist")

    # Check if the gunzip function is used with GunzipParams in the
    middleware file
    with open(middleware_file_path, 'r') as file:
        tree = ast.parse(file.read())

    gunzip_params_used = False
    for node in ast.walk(tree):
        if isinstance(node, ast.Call) and isinstance(node.func, ast.Name
        ) and node.func.id == 'gunzip':
            if len(node.args) == 1:
                arg = node.args[0]
                # Check if the argument passed to gunzip is an instance of
                 GunzipParams
                if isinstance(arg, ast.Name) or (isinstance(arg, ast.
                Attribute) and arg.attr == 'GunzipParams'):
                    gunzip_params_used = True
                    break

    self.assertTrue(gunzip_params_used, "gunzip function in '
    httpcompression.py' does not use a 'GunzipParams' object as a
    parameter")

if __name__ == '__main__':
    unittest.main()
```

# E  EXAMPLE TEST OUTPUTS

Here are the results of running a subset of the custom AST unit tests. The outputs showcase the subtask testing formats and the specificity in unit test function names.

```
    Patch Evaluation Results
================================================================================

Test file: tests/django_refactor/adapt_method_mode.py
Test results: Passed
================================================================================

Test file: tests/salt_refactor/cant-create-test.py
Error: test_ex_cantcreat_isnt_used (cant-create-test.TestSaltExitCodes.
test_ex_cantcreat_isnt_used) ... FAIL
test_ex_cantcreate_in_exitcodes (cant-create-test.TestSaltExitCodes.
test_ex_cantcreate_in_exitcodes) ... ok
test_ex_cantcreate_in_ssh_py_shim (cant-create-test.TestSaltExitCodes.
test_ex_cantcreate_in_ssh_py_shim) ... FAIL
test_ex_cantcreate_is_used (cant-create-test.TestSaltExitCodes.
test_ex_cantcreate_is_used) ... FAIL
test_exitcodes_does_not_have_ex_cantcreat (cant-create-test.
TestSaltExitCodes.test_exitcodes_does_not_have_ex_cantcreat) ... ok
test_ssh_py_shim_does_not_have_ex_cantcreat (cant-create-test.
TestSaltExitCodes.test_ssh_py_shim_does_not_have_ex_cantcreat) ... FAIL
test_ssh_py_shim_does_not_import_exitcodes (cant-create-test.
TestSaltExitCodes.test_ssh_py_shim_does_not_import_exitcodes) ... ok
test_ssh_py_shim_uses_local_ex_cantcreate (cant-create-test.
TestSaltExitCodes.test_ssh_py_shim_uses_local_ex_cantcreate) ... FAIL

======================================================================
FAIL: test_ex_cantcreat_isnt_used (cant-create-test.TestSaltExitCodes.
test_ex_cantcreat_isnt_used)
----------------------------------------------------------------------
Traceback (most recent call last):
  File "/refactor_repos/salt_refactor/task_test/cant-create-test.py", line
    155, in test_ex_cantcreat_isnt_used
    self.assertFalse(ex_cantcreat_found, "salt.defaults.exitcodes.
    EX_CANTCREAT was found in salt/client/ssh/__init__.py, but it should
    not be used.")
AssertionError: True is not false : salt.defaults.exitcodes.EX_CANTCREAT
was found in salt/client/ssh/__init__.py, but it should not be used.
======================================================================
FAIL: test_ex_cantcreate_in_ssh_py_shim (cant-create-test.
TestSaltExitCodes.test_ex_cantcreate_in_ssh_py_shim)
----------------------------------------------------------------------
Traceback (most recent call last):
  File "/refactor_repos/salt_refactor/task_test/cant-create-test.py", line
    39, in test_ex_cantcreate_in_ssh_py_shim
    self.assertTrue(ex_cantcreate_found, f"'EX_CANTCREATE' not found in {
    file_path}")
AssertionError: False is not true : 'EX_CANTCREATE' not found in ../salt/
client/ssh/ssh_py_shim.py
======================================================================
FAIL: test_ex_cantcreate_is_used (cant-create-test.TestSaltExitCodes.
test_ex_cantcreate_is_used)
----------------------------------------------------------------------
Traceback (most recent call last):
  File "/refactor_repos/salt_refactor/task_test/cant-create-test.py", line
    131, in test_ex_cantcreate_is_used
    self.assertTrue(ex_cantcreate_found, "salt.defaults.exitcodes.
    EX_CANTCREATE was not found in salt/client/ssh/__init__.py")
AssertionError: False is not true : salt.defaults.exitcodes.EX_CANTCREATE
 was not found in salt/client/ssh/__init__.py
======================================================================
```

```
FAIL: test_ssh_py_shim_does_not_have_ex_cantcreat (cant-create-test.
TestSaltExitCodes.test_ssh_py_shim_does_not_have_ex_cantcreat)
-------------------------------------------------------------------
Traceback (most recent call last):
  File "/refactor_repos/salt_refactor/task_test/cant-create-test.py", line
    107, in test_ssh_py_shim_does_not_have_ex_cantcreat
      self.assertFalse(ex_cantcreat_found, f"'EX_CANTCREAT' (misspelled)
      found in {file_path}")
AssertionError: True is not false : 'EX_CANTCREAT' (misspelled) found in
../salt/client/ssh/ssh_py_shim.py
===================================================================
FAIL: test_ssh_py_shim_uses_local_ex_cantcreate (cant-create-test.
TestSaltExitCodes.test_ssh_py_shim_uses_local_ex_cantcreate)
-------------------------------------------------------------------
Traceback (most recent call last):
  File "/refactor_repos/salt_refactor/task_test/cant-create-test.py", line
    55, in test_ssh_py_shim_uses_local_ex_cantcreate
      self.assertTrue(ex_cantcreate_used, f"'EX_CANTCREATE' not used in {
      file_path}")
AssertionError: False is not true : 'EX_CANTCREATE' not used in ../salt/
client/ssh/ssh_py_shim.py

-------------------------------------------------------------------
Ran 8 tests in 0.046s
FAILED (failures=5)

===================================================================================

Test file: tests/fastapi_refactor/value-is-a-sequence-test.py
Error: test_compat_file_exists (value-is-a-sequence-test.
TestFastAPICompatUtils.test_compat_file_exists) ... ok
test_import_value_is_a_sequence_in_utils (value-is-a-sequence-test.
TestFastAPICompatUtils.test_import_value_is_a_sequence_in_utils) ... FAIL
test_value_is_a_sequence_function_exists (value-is-a-sequence-test.
TestFastAPICompatUtils.test_value_is_a_sequence_function_exists) ... ok
test_value_is_sequence_function_does_not_exist (value-is-a-sequence-test.
TestFastAPICompatUtils.test_value_is_sequence_function_does_not_exist)
... ok
test_value_is_sequence_function_does_not_exist_in_utils (value-is-a-
sequence-test.TestFastAPICompatUtils.
test_value_is_sequence_function_does_not_exist_in_utils) ... ok

===================================================================
FAIL: test_import_value_is_a_sequence_in_utils (value-is-a-sequence-test.
TestFastAPICompatUtils.test_import_value_is_a_sequence_in_utils)
-------------------------------------------------------------------
Traceback (most recent call last):
  File "/refactor_repos/fastapi_refactor/task_test/value-is-a-sequence-
  test.py", line 75, in test_import_value_is_a_sequence_in_utils
      self.assertTrue(import_found, "'value_is_a_sequence' not imported from
      'fastapi._compat' in dependencies/utils.py")
AssertionError: False is not true : 'value_is_a_sequence' not imported
from 'fastapi._compat' in dependencies/utils.py

-------------------------------------------------------------------
Ran 5 tests in 0.026s
FAILED (failures=1)

===================================================================================

Test file: tests/scrapy_refactor/add-log-parameter-xmliter.py
Test results: Passed
===================================================================================
```

## F State Reconstruction Experiment

We prompt `gpt-4-turbo` with randomly initialized and randomly changed lists of preferences. We generate preferences through this script and simply iteratively prompt with subportions of the json.

```python
import json
import random

def generate_random_preferences(categories, products):
    return {
        category: {
            product: random.choice(["Likes", "Dislikes", "NA"]) for product
            in products[category]
        } for category in categories
    }

def generate_trajectory(initial_prefs, num_actions, categories, products)
:
    actions = ["SetPreference"]
    trajectories = []
    preferences = {cat: dict(initial_prefs[cat]) for cat in initial_prefs}
     # Deep copy to prevent mutation

    trajectory = {"actions": [], "states": {}}
    for i in range(1, num_actions + 1):
        category = random.choice(categories)
        product = random.choice(products[category])
        action_type = random.choice(actions)
        old_preference = preferences[category][product] # Track old
        preference
        new_preference = random.choice(["Likes", "Dislikes", "NA"])
        while new_preference == old_preference: # Ensure the new preference
         is different
            new_preference = random.choice(["Likes", "Dislikes", "NA"])

        details = {
            "action": action_type,
            "category": category,
            "product": product,
            #"old_preference": old_preference,
            "new_preference": new_preference
        }
        preferences[category][product] = new_preference # Update to new
        preference
        trajectory["actions"].append(details)

        # Snapshot of system state after each action
        trajectory["states"][f"Action{i}"] = {cat: dict(preferences[cat])
        for cat in preferences}

    trajectories.append(trajectory)
    return trajectories

def main():
    categories = ["Electronics", "Books", "Clothing", "Garden", "Games"]
    products = {
        "Electronics": ["Laptop", "Smartphone", "Headphones"],
        "Books": ["Novel", "Biography", "Science Fiction"],
        "Clothing": ["Jeans", "T-Shirt", "Jacket"],
        "Garden": ["Shovel", "Lawn Mower", "Gloves"],
        "Games": ["Board Game", "Video Game", "Puzzle"]
    }
    num_initial_states = 50
    trajectories_per_state = 5
```

```
    actions_per_trajectory = 50 # Example, number of actions per
    trajectory

    all_data = []

    for _ in range(num_initial_states):
        initial_prefs = generate_random_preferences(categories, products)
        trajectories = []
        for _ in range(trajectories_per_state):
            trajectory = generate_trajectory(initial_prefs,
            actions_per_trajectory, categories, products)
            trajectories.extend(trajectory) # Add the generated trajectory
            to the list
        all_data.append({
            "initial_preferences": initial_prefs,
            "trajectories": trajectories
        })

    with open('complex_actions_states.json', 'w') as f:
        json.dump(all_data, f, indent=4)

if __name__ == "__main__":
    main()
```

Given these randomly generated actions in json, we prompt `gpt-4-turbo` for 125 random initializations iteratively over 0-50 actions of the generated actions. Figure 7 shows the prompt and expected output. We are not strict with format rules, and allow minor mistakes, however, our parser requires the larger category separations.

---

**Toy Agent Reconstruction Task**

```
Here are your initial preferences on 5 different categories.
Preferences:
{ 'Electronics': { 'Laptop': 'Likes', 'Smartphone': 'Likes', 'Headphones': 'Dislikes' }, 'Books': { 'Novel':
'Dislikes', 'Biography': 'NA', 'Science Fiction': 'Dislikes' }, 'Clothing': { 'Jeans': 'Likes', 'T-Shirt':
'Likes', 'Jacket': 'Likes' }, 'Garden': { 'Shovel': 'Likes', 'Lawn Mower': 'NA', 'Gloves': 'NA' }, 'Games':
{ 'Board Game': 'Likes', 'Video Game': 'Likes', 'Puzzle': 'Likes' } }
Here are the actions in order after that initial state:
Action 1: Electronics - Laptop to 'NA'.
...
Action N: Clothing - T-Shirt to 'NA'.
This is the end of the changes. What is the state of preferences on all categories after the actions? Format
your response EXACTLY how I formatted the input initial preferences state. Preferences:
```
---
```
Desired answer: { 'Electronics': { 'Laptop': 'NA', 'Smartphone': 'Likes', 'Headphones': 'Dislikes' },

'Books': { 'Novel': 'Dislikes', 'Biography': 'NA', 'Science Fiction': 'Dislikes' }, 'Clothing': { 'Jeans':

'Likes', 'T-Shirt': 'NA', 'Jacket': 'Likes' }, 'Garden': { 'Shovel': 'Likes', 'Lawn Mower': 'Dislikes',

'Gloves': 'NA' }, 'Games': { 'Board Game': 'Likes', 'Video Game': 'Dislikes', 'Puzzle': 'Likes' } }
```

Figure 7: Example of a singular instance of the synthetic state construction task.

## G    SIMPLE SINGLE AGENT STATE-AWARE IMPLEMENTATION

As an self-contained example, we have a simple implementation of a state-aware interface contained within a singular agent instance. This state command tracks all it's previous edit commands and concatenates them in a separate section. In practice and for results in the paper, we augment the state cache to relay more information about related edits by integrating parts of previous observations as well.

```
state_command:
  name: state
  code: |
    state() {
      local working_dir="$PWD"
      local open_file_info="${CURRENT_FILE:-n/a}"
      local recent_edits_json='[]'

      if [ -n "$RECENT_EDITS" ]; then
          # Split $RECENT_EDITS into an array of edits
          IFS=' | ' read -r -a edits_array <<< "$RECENT_EDITS"
          declare -A seen_edits

          # Filter out duplicate filename-line_number pairs
          filtered_edits=()
          for edit in "${edits_array[@]}"; do
              filename=$(echo "$edit" | cut -d':' -f1)
              line_number=$(echo "$edit" | cut -d':' -f2)

              # Check if this filename:line_number pair has been seen before
              if [ -z "${seen_edits["$filename:$line_number"]}" ]; then
                  filtered_edits+=("$edit")
                  seen_edits["$filename:$line_number"]=1
              fi
          done

          # Convert the filtered edits into a JSON array
          recent_edits_json=$(printf '%s\n' "${filtered_edits[@]}" | jq -R -
          s -c 'split("\n")')
      fi

      state_json=$(jq -n --arg wd "$working_dir" --arg of "$(realpath "
      $open_file_info")" --argjson re "$recent_edits_json" \
          '{"working_dir": $wd, "open_file": $of, "recent_edits": $re}')

      echo "$state_json"
    }
```

We also, to communicate all the recent edits, append the outputs of this state command in every step template through:

```
  next_step_template: |-
    {observation}
    (Current State: {recent_edits})
    (Open file: {open_file})
    (Current directory: {working_dir})
    bash-$
```

## H  EXAMPLE STATE-AWARE OBJECTIVE UPDATE

In this example excerpt from a trajectory, we are able to see our state-aware agent reason about an external edit (propagated through the state-aware policy) made to the local repository it was generating a concurrent patch for.

```
{
  "role": "assistant",

  "thought":
  "DISCUSSION\nWe have found the `get_encoding_from_headers` function
  calls at lines 622 and 623 in the `test_utils.py` file. Let's
  navigate to line 622 and update the function calls to pass `log=False
  `.\n",
  "action": "goto 622\n",
  "agent": "primary"
},
{
  "role": "user",
  "content":

    "[File Contents Printed, Truncated for Brevity]

    (External Edits:  ['Since your previous action, another user edited
    adapters.py at lines 359:359
    n'])

    (Your Recent Edits: ['Edited utils.py at lines 542:542', 'Edited
    adapters.py at lines 378:378', 'Edited adapters.py at lines
    618:618\\n'])

    (Open file: /requests_refactor/tests/test_utils.py)

    (Current directory: requests_refactor)
    bash-$",
  "agent": "primary"
},
{
  "role": "assistant",
  "thought": "DISCUSSION\nLet's edit the function call at line 623 to
  pass `log=False`. Then, let's view the external edit in
  adapters.py.",

  "action":
  "
  edit 623:623
  assert get_encoding_from_headers(value, log=False) == expected
  end_of_edit
  ",
}
```

## I CONTEXT FLOODING FIGURE

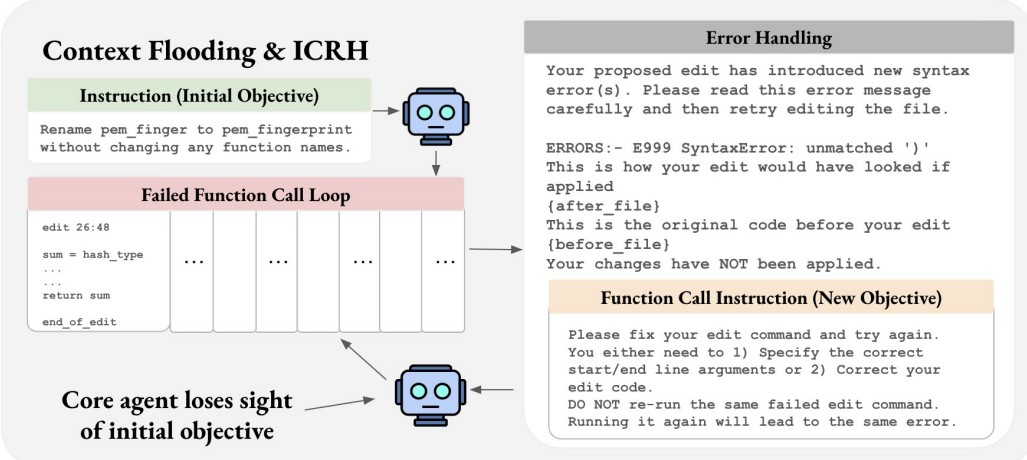

Figure 8: Visual example of a language agent having a failed function call loop showcasing the context flooding and deprioritized objective failure mode.

