# OpenReview forum: "RefactorBench: Evaluating Stateful Reasoning in Language Agents Through Code"
_ICLR.cc/2025/Conference — ICLR 2025 Poster_

### Official Review · Reviewer_vwuT · 2024-10-29

**Soundness:** 3
**Presentation:** 4
**Contribution:** 2
**Rating:** 6
**Confidence:** 4

**Summary:**

The sources present *RefactorBench*, a novel benchmark for evaluating language models' (LMs) capabilities in multi-file code refactoring tasks. The benchmark comprises 100 handcrafted tasks requiring edits across multiple files, with varying instruction sets to assess the models' ability to follow instructions and reason over past actions. The authors demonstrate that current LMs struggle with these complex tasks, particularly due to difficulties tracking past actions and maintaining context. To address this limitation, they propose "state-aware" interfaces that provide LMs with a representation of the current state of the codebase, leading to significant improvements in task performance. Finally, the authors outline potential future directions for research, including developing more efficient context generalization techniques and constructing robust interaction mechanisms for language agents in digital environments.

**Strengths:**

This is a well-written benchmark paper with clear motivation.

1. Filling the gap between existing bug-fixing benchmarks (e.g., SWE-Bench) and the need for real-world developers (e.g., code refactoring). Repo-level code generation benchmarks [1, 2, 3] are necessary.
2. Well-organized and comprehensive literature review for automated software engineering (ASE) systems.
3. Clear data collection procedure. Statistics on RefactorBench tasks, repositories, and AST-based unit tests are included.
4. Clear problem formulation as POMDP. Standardized evaluation with AST-based unit tests. The code is easy to run from anonymous GitHub.
5. An interesting discovery of the SWE-Agent prone to fix bugs instead of refactoring. The error analysis is also comprehensive.
6. An interesting discovery of state awareness's significance in enhancing language agents' capabilities with a synthetic experiment. The figure and flowchart are easy to understand.


[1] Zhang, F., Chen, B., Zhang, Y., Keung, J., Liu, J., Zan, D., ... & Chen, W. (2023, December). RepoCoder: Repository-Level Code Completion Through Iterative Retrieval and Generation. In Proceedings of the 2023 Conference on Empirical Methods in Natural Language Processing (pp. 2471-2484).

[2] Li, B., Wu, W., Tang, Z., Shi, L., Yang, J., Li, J., ... & Chen, K. (2024). Devbench: A comprehensive benchmark for software development. arXiv preprint arXiv:2403.08604.

[3] Liu, Y., Tang, X., Cai, Z., Lu, J., Zhang, Y., Shao, Y., ... & Gerstein, M. (2023). ML-Bench: Large Language Models Leverage Open-source Libraries for Machine Learning Tasks. arXiv e-prints, arXiv-2311.

[4] Huang, Q., Vora, J., Liang, P., & Leskovec, J. (2023). Benchmarking Large Language Models as AI Research Agents. In NeurIPS 2023 Foundation Models for Decision Making Workshop.

**Weaknesses:**

This paper has several weaknesses regarding the evaluation methodologies.
1. The tasks are relatively easy:
    - For most of the tasks given in the anonymous GitHub, I've learned that LLMs cannot even move some code snippets to another file or add additional parameters to one function. Is that correct?
       - In `problems/ansible_refactor/sort-groups-to-group-sort-task.txt`: Rename `sort_groups` to `group_sort` and update all references in the repo.
       - In `problems/ansible_refactor/add-log-parameter-get-group-vars-task.txt`: Add a log parameter to `get_group_vars,` keep the base case as true, and update the rest of the repository calls to have `log=False.`
       - $\ldots$
    - From my experience using Cursor and GitHub Copilot, the observation in this paper is not true. Most refactoring like these can be solved with one click of the mouse.
       - For example, I can modify the original function using the Cursor and open all the relevant files (i.e., Ctrl+F for that specific function). The Cursor will devise an auto-completion suggestion to refactor all usage of this function.
       I am pretty sure that if our language agent is equipped with some retrieval tools and employs a hierarchical structure (i.e., multi-agent) in which each agent updates one file separately, it would largely improve performance and consistency.

2. Bad definition of refactoring:
    - Martin Fowler's "Repetition is the root of all software evil" is a good point. However, I did not see any prompt examples of reducing repetition within the refactor tasks. From my daily experience, a good refactor should reduce the complexity and repetition of a large codebase and increase the portability of modules and functions. It is not about adding a logger to a function or moving some code to `utils.py.`
    - The refactor should not be evaluated using unit tests only. Unlike the bug-fixes scenario in SWE-Bench, the function before refactoring should also pass the unit test. The objective should be, for example, to reduce the length/loops of the initial code after refactoring while still passing the unit test.
    - I don't have a clear idea of how to evaluate the refactoring process, but it is obvious that using a pass rate for unit tests is insufficient. Below are some potential metrics you can choose from (maybe for your next benchmark).
      - *Test Execution Time*: After refactoring, measure if the test suite executes faster.
      - *Linting and Style Compliance*: Assess adherence to coding standards. (This might not be helpful because most code generated by LLM should follow linting compliance).
      - *Lines of Code (LOC)*: While not always directly indicative of quality, a reduction in LOC often reflects simplification.
      - *Function Size*: Track average function length; shorter, well-named functions are generally easier to understand and test.
      - *Coupling Metrics*: Evaluate coupling using Coupling Between Objects (CBO) measures. (I hate the coupling of my code. So, if the LLMs can solve that, it will be really helpful.)
      - *Dependency Analysis*: Check the number and nature of dependencies between modules. Reducing unnecessary dependencies enhances modularity and flexibility in the code.

3. Other frameworks worthy to test on [1, 2, 3]
     - Aider [1] also has a good way of code edits and refactoring using `git diff` [4].
     - Papers such as [2, 5] evaluate their performance on multiple agent frameworks.

[1] P. Gauthier. How aider scored sota 26.3. https://aider.chat/2024/05/22/swe-bench-lite.html. Accessed: 2024-06-05.

[2] Wang, X., Li, B., Song, Y., Xu, F. F., Tang, X., Zhuge, M., ... & Neubig, G. (2024). OpenHands: An Open Platform for AI Software Developers as Generalist Agents. arXiv preprint arXiv:2407.16741.

[3] Xia, C. S., Deng, Y., Dunn, S., & Zhang, L. (2024). Agentless: Demystifying llm-based software engineering agents. arXiv preprint arXiv:2407.01489.

[4] P. Gauthier. Unified diffs make GPT-4 Turbo 3X less lazy. https://aider.chat/2023/12/21/unified-diffs.html. Accessed: 2024-10-31.


[5] Liu, Y., Tang, X., Cai, Z., Lu, J., Zhang, Y., Shao, Y., ... & Gerstein, M. (2023). ML-Bench: Large Language Models Leverage Open-source Libraries for Machine Learning Tasks. arXiv e-prints, arXiv-2311.

**Questions:**

''Repetition is the root of all software evil'' -- Martin Fowler

It is a reference to the software engineering principle of Don't Repeat Yourself (DRY). The idea behind DRY is that repeating code or logic in multiple places can lead to problems.

What is the relationship of this quote with the whole paper?

---

> ### Author Response · Authors · 2024-11-24
>
> Thank you for your review and constructive feedback!
>
> Please refer to the general response for a summary of some other updates. To address your specific concerns and questions:
>
> **1. Evaluations of Refactors (i.e., Using a pass rate of unit tests is insufficient)**
>
> We completely agree that using internal repository unit tests is not enough to verify the correctness of refactors. Therefore, we designed and implemented a set of custom Abstract Syntax Tree (AST) based tests to verify the correctness of refactorings. Through ASTs, one can examine a broad spectrum of correctness conditions: such as the presence of a certain import in a file or presence of a parameter in a specific function call in a specific function. Through manual review, we find that passing all the subtasks in our custom AST-based testing files is an extremely strong method to quantify a refactor being completed.
>
> Because we leverage ASTs in this unique manner, we are able to have evaluations where the relevant areas of code do not pass the custom unit tests before the changes.
>
> Compared to the other approaches listed in your review, we find that ASTs allow for task-specific customizability. Since we are trying to largely evaluate instruction-following for multiple changes across the repository, regardless of the change requested, we do not find non-task-specific heuristics like function size and dependency analysis to be useful in our evaluations. However, we agree that the setups you described could be great ways to expand the scope of the refactoring tasks in the benchmark in future work! Specifically, performance-based refactorings are an area of work we plan on exploring.
>
> We’ve also reworded and reorganized the task construction section to highlight this testing format more.
>
>
> **2. Relevance of the Quote**
>
> The quote from Martin Fowler is a nod to a major goal of refactoring (DRY), while also alluding to one of our takeaways in our analysis section. We find that agents struggle with repetitive action-taking and that repetition is our “root of evil” for creating robust agents.
>
> Here are some task examples within RefactorBench of removing repetition in repositories (their base instructions as well):
>
> - **combine-from-key-to-key-task**:
>   _“Convert the from_key_val_list and the to_key_val_list functions into a singular function key_val_convert(value, to_dict=True) with the behavior that if to_dict is true, it behaves like from_key_val_list and if to_dict is False, it behaves like to_key_val_list. Update the rest of the repo accordingly.”_
>
> - **exactly-n-boto-mod-task.txt**:
>   _“Keep the exactly_n and exactly_1 functions in botomod and just define the functions in boto3mod as the import of imported_exactly_n and imported_exactly_1 so the code isn't as repetitive.”_
>
> - **remove-locale-data-task.txt**:
>   _“tornado._locale_data is only used by tornado.locale. Move the locale data into locale, and remove the original file.”_
>
>
> **3. Task Simplicity**
>
> We agree that adding a human in the loop (e.g., finding the instances manually in Cursor and choosing the files to Cursor Composer) makes some of the tasks simpler for current AI-based code editing tools. However, the specific application we are focusing on in this paper is fully automated program transformations, without any human interventions.
>
> Our experiments show that (with the agentic systems we experimented with) RefactorBench is not easy to complete end-to-end, and efficient exploration in repositories is not a solved problem. The human baseline with a 5-minute time limit is intended to demonstrate this divergence in “simplicity” for humans and agents. While varying levels of assisted systems could show better performance on these tasks, we think RefactorBench is a strong benchmark to get evidence on the performance of such systems.
>
>
> **4. Related Works**
>
> Thanks for those suggestions! We will add the relevant missing references.
>
>
> We greatly appreciate your comments and suggestions. If you have any additional questions or concerns, please let us know. Thank you for your time and consideration.

---

> > ### Comment · Reviewer_vwuT · 2024-11-27
> >
> > Thank you for your feedback. I have revised my score from 5 to 6 to motivate the authors to include more baseline frameworks.

---

### Official Review · Reviewer_2pe1 · 2024-11-01

**Soundness:** 4
**Presentation:** 3
**Contribution:** 3
**Rating:** 6
**Confidence:** 4

**Summary:**

This paper presents RefactorBench, a new benchmark designed to evaluate language agents ability to perform non-bug-fixing refactoring of multi-file code repositories. The authors run baselines and present analysis of common failure modes of existing LM agents on the task. The authors then present a new approach to improving AI agent task performance by directly modeling and conditioning inference on task state.

**Strengths:**

* clear explanation of generation of the benchmark + helpful metrics of the benchmark by proxies for difficulty (number of files / lines, instruction detail). Many questions can be explored with this dataset by using the different degrees of detail of NL insn, small-granularity feedback from the AST unit tests, and complexity of large repositories
* clean formulation of the task as a POMDP well-addresses the identified failure modes and motivates the state-aware method
* compelling analysis of the results, with proper side experiments and task-related ablations to motivate state-aware agents
* refactoring is an interesting task, as prior works have focused on bug-fixing refactors. The insight that refactoring often necessitates "broken" intermediate states on the way to task completion poses an interesting research question for agentic frameworks.
* representing the "state" as NL keeps the framework flexible and fairly interpretable

**Weaknesses:**

* the main distinction from SWE-bench is described as "handcrafted and underrepresented multi-file refactoring tasks", but SWE-bench does include some tasks that resemble "refactoring" in the form of feature requests.
* I worry that this is a benchmark that would soon be saturated, but I would not necessarily consider that a reason for rejection. Additionally, the contributions of this paper in the form of analysis make this point less relevant.
* I would have liked to see more analysis of why state is important specifically for refactoring. Can we get some qualitative analysis of what sorts of subtasks are done better with state modeling? What are some failure cases in the NL description of the state?

**Questions:**

* how did you choose the repositories to do refactoring on?
* Why do you think that refactoring now causes LM agents to edit the wrong files, while debugging LM agents rather consistently do? Does it have to do with the nature of refactoring being correct --> intermediately broken --> correct but refactored?

---

> ### Author Response · Authors · 2024-11-24
>
> Thank you for your detailed and constructive feedback on our work!
>
> Please refer to the general response for a summary of some other updates. To address your specific questions and concerns:
>
> 1. **Choosing Repositories**
>    We chose repositories by generally finding highly starred Python repositories. As test cases were handcrafted, we found repository choice rather important in the ability to create varying levels of difficulty in test cases (need multi-file complexity). As such, we find that the repositories we finalized on often relate to interactive Python frameworks, rather than mostly static libraries.
>
> 2. **Reasoning Behind Edit Localization Failures**
>
>    Here are two of the strongest lines of reasoning that we believe are major causes of the failure mode of edit localization:
>
>    - We know that language agents struggle with the composition of actions and understanding what the cumulative effects are for divergence from some initial state (Section 6 and Figure 4). For a task like refactoring that necessitates smaller edits in multiple files, keeping track of such information is difficult for the language agent. For instance, if one needs to update an import statement in 12 files, keeping track of the files already updated is extremely important. In practice, we find that agents don’t complete the refactors and aren’t able to localize the relevant X locations, but can localize a single location well (struggling with multiple objectives). However, our state-aware implementation does theoretically stop this failure, but the failure mode still persists, just at a lower rate, which hints that there are some more fundamental issues than just action tracking.
>
>    - We also agree with your point and think that the task style of refactors going from (correct → intermediately broken → correct but refactored) is difficult for core LMs not necessarily trained to function in the long horizon for this task style. This sort of reasoning has been shown in previous work where the authors tackled training progress models for situations with unclear reward [1]. However, our horizon length for refactoring is not comparable to the tested environments and thus we believe this is a contributing factor rather than the sole cause.
>
>    [1] Bruce, J., Anand, A., Mazoure, B., & Fergus, R. (2023). *Learning about progress from experts.* In *The Eleventh International Conference on Learning Representations.*
>
> 3. **Why State is Important**
>    As subtasks are often very specific to the overall refactoring tasks, we generally leaned away from finding specific trends within subtasks in the paper. However, there are multiple takeaways of more subtasks being completed and a few qualitative assessments we can make about those tasks:
>
>    - **The agent is reaching more files:** Files/changes that are not the initial step in the refactor are being tackled, demonstrating more long-term coherence.
>    - **The agent is coherently making edits based on previous edits.** Many of the subtask tests are dependent on previous tests (e.g., an import would not be updated with a new filename if some filename wasn’t altered to begin with) and this shows that the agent is focusing on the initial goal better, improving on the issue of objective misalignment.
>    - **Failure Cases of State:** A consistent issue we find with our agents is that making changes inside a core code for a repository and then updating relevant tests in a whole different subsection of the repository is difficult. We hypothesize that having a stronger, compact representation of the current changes from initialization would help in updating far, but related files like in-repository tests. Additionally, on the agent side, exploring passing in multiple files to the core LM (similar to humans working with multiple monitors or split tabs in their IDE) could potentially aid in this failure mode.
>
> We hope these clarifications address your questions! If you have any additional questions or concerns, please let us know. Thank you for your time and consideration.

---

### Official Review · Reviewer_VXXQ · 2024-11-04

**Soundness:** 3
**Presentation:** 2
**Contribution:** 3
**Rating:** 8
**Confidence:** 3

**Summary:**

RefactorBench introduces a new benchmark for evaluating LM agents on performing a variety of multi-file "refactoring" tasks. For instance, one task may involve adding new arguments to a class with sensible default values and updating dependent / affected components throughout the code base. There seem to be other diverse tasks. The authors show that these tasks, while seemingly straightforward for human programmers (most being solvable within 5 minutes by humans), can still be a challenge for existing agents when evaluated on AST-based unit tests against reference solutions. They provide a few evaluation settings, such as having a `lazy`, `standard`, and  `descriptive` version of each task and provide analysis of the dataset and model performance. Finally, state-aware interfaces for LM agents are proposed as an improvement over a simple SWE-agent baseline agent, which includes providing LMs a summary of code changes made overtime, that is demonstrated to improve performance.

**Strengths:**

- RefactorBench introduces a practical setting that remains relatively difficult for LM agents.
- The data quality seems fairly high based on my manual inspection.
- The three-tiered instruction approach (lazy, standard, descriptive) offers valuable insights into agent capabilities.
- The proposed improvements to SWE-agent and agent architectures in general represent meaningful advances.
- Provides thorough analysis.

**Weaknesses:**

- The term `pseudo-task` is used in the abstract but doesn't seem to be used or elaborated on explicitly in the main text. I recommend using a more common term in the abstract, especially if it isn't defined elsewhere.
- Tractability is first used in Section 3.1 but is not defined until a bit later in section 3.2. I recommend pointing to this in the text in 3.1 when it is first used or cite a work that introduces this term. It doesn't seem conventional enough to use without definition.
- Figure 1 lacks substance and could be more much informative.
- With 100 instances the benchmark remains a bit small - however it may be understandable provided that there may a fair bit of human labor involved in creating instances.
- The lack of a train or, more importantly, a development set may limit the useful lifetime of this benchmark. Test-split only benchmarks encourage overfitting when they require some upfront system design (like agent systems). This may also have been a problem in the development of the agent systems presented in this paper.
- The writing is acceptable but could be slightly simplified or reorganized for greater clarity.

**Questions:**

- Line 162: can you clarify what you mean by design principle?
- Seems to be a mistake on line 415: Perhaps you mean a 40.4% relative improvement in performance?

---

> ### Author Response · Authors · 2024-11-24
>
> Thank you so much for your review!
>
> Please refer to the general response for a summary of some other updates. To address your specific questions and concerns:
>
> **Design Principle:**
>
> - We use design principle as a type of refactor described in Martin Fowler’s “Refactoring.” In practice, multiple design principles are used per task, but we often find that the localization in Step 1 ends up leading towards a refactor using between 1-3 styles of refactors, with one general design principle being our classification.
>
> - Here’s a comprehensive list: [https://refactoring.com/catalog/](https://refactoring.com/catalog/)
>
> **Definitions and Line 415:**
>
> - Fixed!
>
> **Lack of a Train Set:**
>
> - We initially thought about a train split but reached the conclusion that refactoring-style code changes would follow similar results to SWE-bench’s train set, where fine-tuning on before-and-after git diff pairs had minimal improvement. Additionally, the automated scripting possible for SWE-bench’s train set outsizes our capabilities to create handcrafted refactors for training.
>
> - As a side note, we expect fine-tuning on custom agent trajectories to be more effective than the standard train split in improving LM agents.
>
> We have made some changes to the writing/structuring of the work. Please take a look and let us know if there are any more specific concerns.
>
> We greatly appreciate your comments and suggestions. If you have any additional questions or concerns, please let us know. Thank you for your time and consideration.

---

### Official Review · Reviewer_LgvP · 2024-11-07

**Soundness:** 3
**Presentation:** 3
**Contribution:** 3
**Rating:** 6
**Confidence:** 4

**Summary:**

This paper introduces a novel benchmark, RefactorBench, designed to evaluate the limitations of LM agents in complex, multi-file code refactoring tasks within open-source repositories. RefactorBench consists of 100 handcrafted tasks, each defined by 3 natural language instructions that vary in specificity, requiring agents to explore dependencies across multiple files to complete tasks successfully. Initial experiments demonstrate that current LM agents struggle  with these compositional tasks, solving only 18% compared to human developers, who achieve 87% under short time constraints. Through trajectory analysis, the authors identify distinct failure modes, particularly the agents' difficulty in tracking past actions. To improve this, the authors propose to adapt baseline agents to condition on state representations which improves performance by 40.4%. The study positions RefactorBench as a critical tool for advancing the  development of LM agents in code-related environments.

**Strengths:**

* RefactorBench is a novel benchmark to evaluate LLMs in the complex multi-file code refactoring tasks, which can isolate unique language agent behaviors compared with similar benchmarks like SWE-bench.
* The benchmark construction process is technically sound, with unit tests included to ensure rigorous correctness assessment. The experiments provide insightful analysis of how current LM agents perform on this new task. Furthermore, rather than simply discussing the results, the authors go a step further by proposing enhancements based on their findings, achieving a 40.4% improvement in performance over baseline agents.

**Weaknesses:**

A key weakness of this work is its narrow focus on GPT-4o and SWE-Agent for evaluation, omitting other closed models like Claude, as well as open-source models and other LM agent frameworks. For a benchmark paper, evaluating a broader range of baselines would provide a more comprehensive view of existing LLM performance and help new models or LM agents identify their relative strengths and weaknesses. Additionally, a wider evaluation would allow for assessing whether the observed results generalize across different models, strengthening the benchmark’s robustness and relevance to the broader field.

**Questions:**

See the above weakness.

---

> ### Author Response · Authors · 2024-11-24
>
> Thank you for your review!
>
> Please refer to the general response for a summary of some other updates. To address your specific concerns:
>
> **Evaluations on Other Models and Agent Frameworks**
>
> - Due to cost limitations and rate limits, we’ve only added evaluations of SWE-agent with Claude 3.5 Sonnet on the descriptive instruction set, solving 35% of instructions. We believe our results on state awareness are isolated to the agent scaffold itself, and we would have similar results across model families, something seen in many previous works in agent benchmarks. Additionally, we find the results of performance on instruction sets to clearly follow the amount of description given in the instruction text.
>
> - As such, to allocate our API resources wisely, in regards to other agent frameworks, we plan on next evaluating OpenHands’ recently released CodeActv2.1 agent that scored SOTA on SWE-Bench Verified soon (still batching smaller runs due to daily rate limits). We also plan on publicly adding external agent evaluations after the anonymity period for people to test their proprietary scaffolds on RefactorBench.
>
> If you have any additional questions or concerns, please let us know. Thank you for your time and consideration.

---

> > ### Comment · Reviewer_LgvP · 2024-11-27
> > **Official Comment by Reviewer**
> >
> > Thanks for the response. I would keep my rating as my concern of limited evaluation is not fully addressed.

---

### Author Response · Authors · 2024-11-24
**General Response**

We truly appreciate all the reviewers' time and feedback on RefactorBench! We have incorporated the rebuttal content into our paper revision to help answer the reviewers’ questions and concerns.

RefactorBench aims to show that truly diverse evaluations for agents are difficult, and careful task design can reveal a variety of new failure modes and inspire new methods (e.g., state-awareness) to tackle them.

To summarize our contributions:
- **RefactorBench**: A high-quality, simple, and comprehensive benchmark that focuses on multi-file refactoring in large Python repositories.
- **Baselines & Analysis**: We provide baseline performance and insights for a prompt-adjusted agent with gpt-4o and Claude 3.5 Sonnet. We show analysis on three novel major failure modes and describe initial agent overfitting to bug fixing.
- **State-awareness**: We tackle a hypothesized issue related to recall in language agents by isolating it in a synthetic scenario and then adding a novel state update method. We also generalize our solution to an environment-wide level and discuss multi-agent implications of state-aware policies.
- **Future Directions**: We outline two major fields for improvement in language agents and describe specific avenues that seem fruitful based on our work and related works.

In response to the reviewers’ questions and feedback, we have made modifications:
- **Clarified Language and Minor Mistakes**:  We updated language in the paper that reviewers found confusing, and changed our evaluation descriptions to more clearly explain our contributions.
- **Rerun Baselines**:  We reran all baselines (fixing an issue where some runs stopped and restarted, wasting a time step) and found minor increases in task performance. We also plan on adding more baselines after the anonymity period for external agent evaluations.
- **Cited More Relevant Works**:  We expanded our references to include more relevant related works.

We thank the reviewers for their feedback, and we’re happy to answer any further questions.

---

### Meta-Review · Area_Chair_KLx8 · 2024-12-16

**Metareview:**

This paper presents RefactorBench, a novel benchmark for assessing language model agents’ performance in complex, multi-file code refactoring tasks within open-source repositories. The work analyzes baseline models and propose a state-aware approach that significantly improves performance, though the benchmark remains challenging.

This benchmark is valuable. It focuses on an under-explored task of multi-file refactoring with a sound evaluation framework, RefactorBench provides valuable insights into agent capabilities and areas for future improvement.

**Additional Comments On Reviewer Discussion:**

The authors clarified some unclear statements in the paper durning the rebuttal period.

---

### Decision · Program_Chairs · 2025-01-22

Accept (Poster)